# The influence of training sample size on the accuracy of deep learning models for the prediction of soil properties with NIR spectroscopy data

Wartini Ng[1], Budiman Minasny[1], Wanderson de Sousa Mendes[2], José A.M.Demattê [2],

[1] School of Life and Environmental Sciences & Sydney Institute of Agriculture, The University of Sydney, NSW, Australia

[2] Department of Soil Science, "Luiz de Queiroz" College of Agriculture, University of São Paulo, Av. Pádua Dias 11, Portal Box 9, Piracicaba, São Paulo state Code 13418-900, Brazil

*Correspondence to*: Wartini Ng (wartini.ng@sydney.edu.au)

## Abstract

The number of samples used in the calibration dataset affects the quality of the generated predictive models using visible, near and shortwave infrared (VIS-NIR-SWIR) spectroscopy for soil attributes. Recently, the convolutional neural network (CNN) is regarded as a highly accurate model for predicting soil properties on a large database. However, it has not been ascertained yet how large the sample size should be for CNN model to be effective. This paper investigates the effect of training sample size on the accuracy of deep learning and machine learning models. It aims at providing an estimate of how much calibration samples are needed to improve the model performance of soil properties predictions with CNN as compared to conventional machine learning models. In addition, this paper also looks at a way to interpret the CNN models, which are commonly labelled as black box. It is hypothesized that the performance of machine learning models will increase with an increasing number of training samples, but it will plateau when it reached a certain number, while the performance of CNN will keep improving. The performances of two machine learning models (Partial least squares regression (PLSR) and Cubist) are compared against the CNN model. A VIS-NIR-SWIR spectral library from Brazil containing 4251 unique sites, with averages of 2-3 samples per depth (a total of 12,044 samples), was divided into calibration (3188 sites) and validation (1063 sites) sets. A subset of the calibration dataset was then created to represent smaller calibration dataset ranging from 125, 300, 500, 1000, 1500, 2000, 2500 and 2700 unique sites, or equivalent to sample size approximately 350, 840, 1400, 2800, 4200, 5600, 7000, and 7650. All three models (PLSR, Cubist, and CNN models) were generated for each sample size of the unique sites for the prediction of five different soil properties, i.e. cation exchange capacity, organic matter, sand, silt and clay content. These calibration subset sampling processes and modelling were repeated ten times to provide a better representation of the model performances. Learning curves showed that the accuracy increased with an increasing number of training sample. At a lower number of

samples (<1000), PLSR and Cubist performed better than CNN. The performance of CNN outweighed the PLSR and Cubist
model at a sample size of 1500 and 1800 respectively. It can be recommended that deep learning is most efficient for spectral
modelling for sample size above 2000. The accuracy of the PLSR and Cubist model seems to reach a plateau above sample
size of 4200 and 5000, respectively, while the accuracy of CNN has not plateaued. A sensitivity analysis of the CNN model
demonstrated the ability to determine important wavelengths region that affected the predictions of various soil attributes.
***Keywords:*** convolutional neural network, deep learning, machine learning, infrared spectroscopy, soil properties, soil analysis

## 1. Introduction

There has been an increasing demand for a rapid and cost-effective method as an alternative for conventional laboratory soil analysis. Visible, near and shortwave infrared (VIS-NIR-SWIR) spectroscopy has been proposed to be used as an alternative tool for soil analysis for the last few decades (Bendor and Banin, 1995;Shepherd and Walsh, 2002;Stenberg et al., 2010). This method enables the simultaneous prediction of various properties and has non-destructive characteristics.

Various machine learning models, such as Partial Least Squares Regression (PLSR), Cubist, random forest, and support vector machines had been utilized to model spectroscopy data. However, the performances of these regression models are dependent on the spectra pre-processing methods (Rinnan et al., 2009), as well as the size and representativeness of the calibration samples (Kuang and Mouazen, 2012;Ng et al., 2018). Different orders and combinations of the spectra pre-processing methods, developed to remove artefact in the spectral signal, will result in various model performances. Furthermore, the spectra pre-processing techniques developed for a particular dataset might not work for a different dataset. Better generalization can be made by training the model in a larger dataset. However, several studies demonstrated that the performance of the machine learning model did not increase significantly or even plateaued as the calibration sample size increased (Figueroa et al., 2012;Ramirez-Lopez et al., 2014;Ng et al., 2018).

Advances in artificial intelligence, such as deep learning enable the possibility of extracting features from data without hand-engineered features (LeCun et al., 2015), such as pre-processing. Various deep learning convolutional neural network (CNN) model (AlexNet, VGGnet, GoogLeNet, ResNet), had been developed and trained on large volumes of data, which included over 10 million image data (Krizhevsky et al., 2012;Simonyan and Zisserman, 2014;Szegedy et al., 2015;He et al., 2016). Although CNN often deals with images as input data, it has recently been successfully applied to vibrational and reflectance spectroscopy (Acquarelli et al., 2017;Cui and Fearn, 2018;Liu et al., 2018;Ng et al., 2019;Padarian et al., 2019). Acquarelli et al. (2017) found that the CNN based model outperformed other models (Partial Least Square – Least Discriminant Analysis, logistic regression and k-nearest neighbour) for the classification of various vibrational spectroscopy data. CNN also has recently been successfully utilized for regression modelling using reflectance spectroscopy data (Cui and Fearn, 2018;Liu et al., 2018;Ng et al., 2019;Padarian et al., 2019). Cui and Fearn (2018) compared the performance of CNN and PLSR to predict protein and ash content of wheat kernels and wheat flour from the NIR-SWIR spectra data with calibration sample size ranging from 415 – 6,987. Liu et al. (2018) developed one-dimensional CNN model using VIS-NIR-SWIR spectra data to predict soil clay content with a calibration sample size of 16,000. Other studies had shown that CNN model had the capability to outperform PLSR and Cubist model for the prediction of various soil properties using VIS-NIR-SWIR (Ng et al., 2019;Padarian et al., 2019), mid-infrared (MIR) and combined VIS-NIR-SWIR with MIR spectra (Ng et al., 2019) with a calibration sample size greater than 10,000.

Deep learning such as CNN was developed to handle a large amount of data (millions of images), and clearly soil spectra
libraries these days are not that large yet. For example, a recent study used deep learning on 135 soil samples (Chen et al.,
2018). The advantage of using CNN on such a small number of samples is uncertain. A recent review on spectroscopy showed
that there were several studies where deep learning was used with a small calibration sample size (Yang et al., 2019). The
review indicated that increased calibration samples should further improve the calibration performance, however, there is no
guideline how much improvement can be expected and what is the minimum number of samples for it to be effective.
Strategy to select adequate calibration set in terms of representativeness and size is vital in obtaining a model with good
generalization ability. Although various sampling algorithms (Kennard-Stone, conditioned Latin Hypercube sampling, k-
means clustering) to select representative samples have been explored (Ramirez-Lopez et al., 2014;Ng et al., 2018), the
question of how much samples are needed for the CNN model to perform better than machine learning models for spectroscopy
data has yet to be determined. It is commonly depicted and hypothesized in a learning curve that as more data are available,
CNN will perform better compared to traditional machine learning models (Mahapatra, 2018) (see Figure 1). Machine learning
models tend to reach a plateau or show marginal improvement with an increasing amount of data as the model has limited
complexity to deal with an increasing amount of data (Zhu et al., 2016).
Thus, the purpose of this study is to assess the amount of calibration data needed for the CNN model to perform better than
machine learning models. PLSR and Cubist are chosen as the representatives of the machine learning models which had been
found to perform well in soil spectra data (e.g., Dangal et al. (2019)). In addition, to be able to predict soil properties accurately,
we need to understand and interpret how a CNN model can predict soil properties from spectra. Specifically, this paper presents
the following specific contributions:

-   testing the idea that common machine learning models with reach a plateau in accuracy with an increasing number of

calibration samples,

-   establishing the number of calibration samples required for deep learning to be effective for VIS-NIR-SWIR spectra

data,

-   establishing how much improvement in accuracy is achieved the number of calibration sample for deep learning and

machine learning models is increased, and

-   demonstrating how to interpret deep learning model using a sensitivity analysis.

**2.   Materials and Methods**
**2.1.    Dataset and chemical analysis**
This dataset comprises of 12,044 soil samples from 4,251 unique sites. The soil samples, collected from several regions of
Brazil, i.e., states of Sao Paulo, Minas Gerais, Goias, and Mato Grosso do Sul. This dataset is part of The Brazilian Soil Spectral
Library and extracted from Terra et al. (2018) and Bellinaso et al. (2010). The soils were derived mostly from basalt (volcanic
rock) and sedimentary ones (sandstone). Each site has up to seven samples measurements from the surface up to 1 m depth.
The measured properties include soil texture (sand, silt, and clay), organic matter (OM) content and cation exchange capacity
(CEC). The soil particle size was quantified by the pipette method, as described in Donagema et al. (2011). The method consists
of using a 0.1 M NaOH solution as dispersing agent under high-speed mechanical stirring during 10 min. Then, the sand
fraction was separated by sieving and the clay portion by sedimentation. The silt was quantified based on pre- and post-
difference. Organic carbon (OC) was determined by the Walkley and Black method (Walkley and Black, 1934), in which OC
was oxidised using $K_2Cr_2O_7$ in a wet environment and then measured by titration with 0.1 M ammonium iron sulphate. After
that, the organic matter (OM) was calculated by multiplying the OC quantified per the Van Bemmelen factor of 1.724. As
described in Donagemma et al. (2011), a 1 M KCl solution was used to extracted aluminium, exchangeable calcium and
magnesium. The atomic absorption spectrophotometry was used to quantify Ca and Mg concentrations. Aluminium
concentration was determined by titrating with 0.025 M NaOH. Potassium and phosphorus contents were extracted using
Mehlich-1 (0.05 M HCl with 0.0125 M $H_2SO_4$) solution. The concentration of P was quantified by colourimetry and the K
concentration by flame photometry. Afterwards, CEC was determined as the sum of exchangeable cations. The descriptive
statistics of the soil properties measured are included in Table 1.
**2.2.    Spectral measurements**
The VIS-NIR-SWIR spectra of the soil samples were obtained with FieldSpec3 spectroradiometer (Analytical Spectral
Devices, Boulder, Colorado) with a spectral range of visible to shortwave infrared (350 – 2500 nm) and spectral resolution of
1 nm from 350 to 700 nm, 3 nm from 700 to 1400 nm, and 10 nm from 1400 to 2500 nm. The sensor scanned an area of
approximately 2 $cm^2$, and a light source was provided by two external 50-W halogen lamps. These lamps were positioned at a
distance of 35 cm from the sample (non-collimated rays and a zenithal angle of 30°) with an angle of 90° between them. A
Spectralon (Labsphere Inc., North Sutton, NH) standard white plate was scanned every 20 min during calibration. The samples
were oven-dried at 45°C for 48 hours before being ground and sieved ≤ 2 mm. The sample was distributed homogeneously in
Petri dishes for spectra measurement. Three replicates (involving a 180° turn of the Petri dish) were obtained for each sample.
Each spectrum was averaged from 100 readings over 10 s.
**2.3.    Training and validation**
To better represent the soil distribution, we split and subset the data based on sites. The dataset is first randomly split into 75%
calibration (3188 sites) and 25% validation (1063 sites) based on the unique sites.
From the calibration dataset, smaller sample sizes ranging from 125, 300, 500, 1000, 1500, 2000, 2500 and 2700 unique sites
were created, which is equivalent to a sample size of approximately 350, 840, 1400, 2800, 4200, 5600, 7000, and 7650. Better
representations of model performances were provided by ten replicates of these sizes. Each sampling for the same number of
sites could generate a slightly different number of samples since the number of measurements varied from one site to another.
However, the model performance was evaluated on the common validation dataset using a total of 1063 sites (sample size N
= 3017). Thus, we create a learning curve of the accuracy of the models of the validation dataset as a function of the number
of calibration samples.

## 2.4.    Chemometrics model

Prior to the development of machine learning models (PLSR and Cubist), the spectra data were subjected to some pre-
processing methods: (i) conversion to absorbance followed by (ii) Savitzky - Golay smoothing filter with window size of 11
and second-order polynomial (Savitzky and Golay, 1964), (iii) spectral trimming to discard region that has low signal to noise
ratio (<500 nm and between 2450 – 2500 nm)  and (iv) standard-normal-variate (SNV) transformation (Barnes et al., 1989).
For the CNN model, the spectra were only normalized with SNV before being fed into the model. Our previous research (Ng
et al., 2019) found that CNN has its own filtering algorithm that made pre-processing not necessary. This filtering approach
will be discussed in the results section.

### 2.4.1.    PLSR model

PLSR is one of the standard and most commonly used models with the spectroscopy data. It is a linear chemometric regression
model that projects spectra data into latent variables that explain the variances within the spectra data and the response variables
(Wold et al., 1983). The optimal number of latent variables used in the PLSR regression that resulted in the smallest root mean
square error (RMSE) using the cross-validation approach was used to create the models. PLSR was implemented in the R
statistical software (R Core Team, 2019)  using the "pls" package (Mevik et al., 2018).

### 2.4.2.    Cubist model

Cubist is a rule-based data mining model, which is an extension of the M5 model tree by Quinlan (1993). Cubist has been used
successfully in soil spectroscopy studies and in many cases found to perform better than PLSR and other machine learning
models (Dangal et al., 2019). Cubist creates one or more rules, in which if the rules are met, a certain linear model can be
utilized to predict the target task.  The model was evaluated using the "Cubist" package (Kuhn and Quinlan, 2018) in R.

### 2.4.3.    CNN model

The CNN model is composed of three types of layers: convolutional, pooling and fully-connected layer. The convolutional
layer extracts feature from the inputs, the pooling layer reduces the dimensionality of the input feature, and the fully connected
layer connects the outputs from previous layers to the desired target outputs. The CNN model utilized in this study was derived
from our previous study (Ng et al., 2019), where the spectra data were fed into the model as a one-dimensional data. The
architecture of the CNN model is included in Table 2 and Figure 2. Some of the layers within the network are shared to enable
simultaneous output predictions.
The CNN model was trained with an initial learning rate of 0.001 and Adam optimizer (Kingma and Ba, 2014). The network
was trained a batch size of 50, and a maximum epoch of 200. For model optimization purposes, the calibration data is further
divided into 75% train and 25% test set. Dropout, early stopping and reduced learning rates are used as a regularization
technique to prevent network overfitting. For further details of the CNN model, the reader is referred to Ng et al. (2019). The
CNN model was implemented in Python (v3.5.1; Python Software Foundation, 2017) using Keras library (v2.1.2; Chollet,
2015) and Tensorflow (v1.4.1; Abadi et al., 2015) backend.
All the model performances are compared in terms of coefficient of determination ($R^2$), and the root mean square error
(RMSE), bias and ratio of performance to inter-quartile distance (RPIQ) values based on the validation dataset. Generally,
larger values of $R^2$ and RPIQ and smaller bias and RMSE indicate better model performance.

## 2.5.    Sensitivity analysis: evaluating important wavelengths

To uncover how CNN predicts different soil properties, a sensitivity analysis was conducted to assess the importance of each
wavelength in contributing to predictions. Evaluating the sensitivity of the model can be done in several ways, for example,
Cui and Fearn (2018) calculated the sensitivity of a CNN model for NIR by taking a numerical partial derivative of the output
with respect to each wavelength. For wavelength $i$, the sensitivity $S$ was calculated as:

$$S_i = \frac{f(X_1, \ldots, X_i + \varepsilon, \ldots, X_n) - f(X)}{\varepsilon} \qquad \text{(Eq. 1)}$$

where $X$ is the reflectance spectra, and $f(X)$ is the CNN prediction using the spectra, $\varepsilon$ is a small number. The idea is that if
wavelength $i$ has an important contribution to the prediction, a small perturbation to the reflectance value will create a large
change in the prediction.
In our previous study (Ng et al., 2019), we calculated the sensitivity as a function of the variance of the model for each window
of spectra. Here, we calculate the sensitivity based on the variance principle as an alternative approach:

$$S_i = \frac{Var\left(f(X_1, \ldots, X_i, \ldots, X_n) - f(\overline{X})\right)}{Var\ (Y)} \qquad \text{(Eq. 2)}$$

Where $Var$ is the variation calculation, $f(X_1, \ldots, X_i, \ldots, X_n)$ is the prediction of spectra due to variation in wavelength $i$ with
other wavelengths held constant at their mean values, and $f(\overline{X})$ is the prediction value using the mean values of the spectra
and $Y$ is the observed values of the target variable. In essence, we calculated how the model varied in comparison to the
observations as a function of wavelength.
The current sensitivity analysis (Eq. 2) considers the actual variance of the data for a better approximation of wavelengths
sensitivity. To calculate the variance sensitivity, two new data frames are created. The first data frame contains data which is
the average of all the validation spectra data ($\overline{X}$) and the second contains modified average spectra data ($\overline{X}_i$), in which some
of the average measurements are replaced with the actual spectral reflectance at a wavelength width of 5 nm.
The illustrations of the process of deriving new data frames are included in Figure 6. Both data frames are then fed into the
pre-trained CNN model ($f()$). The variance between the average and modified average spectra are then compared to the actual
variance of the target properties as a measure of the model sensitivity (Eq. 2).
**3.   Results**
**3.1.      VIS-NIR-SWIR spectral characteristics**
Large variability within the soil properties and texture could potentially influence the soil spectral characteristics (shown in
Figure 3). In general, there was an increase in reflectance between 400 - 1000 nm, with several prominent absorption features
at 1400, 1900 and 2200nm. Absorption features in the VIS-NIR (400 - 1000 nm)  which is related to iron oxides, such as
haematite ($Fe_2O_3$) and goethite (FeOOH) (Clark, 1999). Absorption near 1400 nm is associated with the first overtone of an
O-H stretch vibration of water or metal-O-H vibration, while absorption is 1900 nm is combination vibrations of water related
to H-O-H bend and O-H stretch (Viscarra Rossel et al., 2009). Absorption in the 2100-2400 nm region is related to the
combination vibrations of minerals. Generally, spectra that have a higher clay content would show smaller reflectance (greater
absorption) values in comparison to those with lower clay content. The representative samples of the VIS-NIR-SWIR spectra
before and after pre-processing were included in Figure 3.
**3.2.      Visualization of the spectra within CNN model**
An attempt to take a look at what the CNN model actually learns was conducted. As the raw reflectance spectrum was fed into
the CNN model, it passed through a convolutional layer which extracted information from the spectra. Filters from the first
two convolutional layers were included in Figure 4. Though only raw spectra were fed into the CNN model, we could see that

the spectra underwent some spectra pre-processing within each filters of the layers. Some of the filters shown in the first convolution layer looked like the input spectra pattern (filter #3, 4 and 10), and some of them mimicked like transformation pattern: absorbance (filter #1, 5, 6, 7, 9, 13 and 16) and derivatives (filter # 2, 8, 11, 12, 14 and 15). The spectrum became smoother when they passed through the second convolutional layer, where some filters only accentuated certain peaks (Figure 4).

### 3.3. Prediction of soil properties and model comparison

The model performances for the validation dataset using the full calibration data ($n_{site}$= 3188, N=9027) for various soil properties and chemometrics model were presented in Table 3. CNN model outperformed both Cubist and PLSR model (in terms of higher $R^2$ and RPIQ and lower RMSE).

The performance achieved using the CNN model with the prediction of sand ($R^2$= 0.85; RPIQ =1.52), silt ($R^2$=0.58; RPIQ =0.75), clay ($R^2$=0.86; RPIQ =1.05), organic matter ($R^2$=0.69; RPIQ =0.91) and CEC ($R^2$=0.68; RPIQ =0.69). Both the PLSR and Cubist had similar performance for the prediction of the various properties. PLSR model achieved $R^2$ of 0.79, 0.47, 0.80, 0.48 and 0.52, and RPIQ of 1.29, 0.67, 0.87, 0.70, and 0.57 for the prediction of sand, silt, clay, organic matter, and CEC respectively. Meanwhile, Cubist model achieved $R^2$ of 0.78, 0.45, 0.81, 0.54 and 0.52 and RPIQ of 1.19, 0.67, 0.92, 0.70 and 0.59 for the prediction of sand, silt, clay, organic matter, and CEC respectively. Nonetheless, on some cases, the CNN model prediction yielded higher bias on the prediction of some soil properties, such as OM and CEC (bias = -0.11 and -0.76 respectively), than PLSR model (bias = 0.04 and -0.17) for the same properties. The Cubist model yielded bias of -0.22 and -0.17 for the prediction of OM and CEC respectively.

Among all the properties predicted, the sand and clay content showed the best performance with $R^2$ values greater than 0.75 regardless of the types of model used ranging from (0.78 – 0.85 and 0.8 – 0.86) respectively. This finding is in agreement with the ones from Demattê et al. (2016), who observed good predictions for sand and clay content with $R^2$ of 0.86 and 0.85. Pinheiro et al. (2017) reported the prediction accuracy of 0.62 and 0.78 for the sand and clay content, respectively. The low performance of the silt predicted can be linked to error associated with the laboratory analysis method, where the silt content is derived from the difference of the soil mass after the sand and clay content are determined. The prediction for OM content in our study ranges from $R^2$ of 0.48 – 0.69. Shibusawa et al. (2001) reported $R^2$ of 0.65 for the prediction of OM using slightly different wavelength region (400-2400nm). Our prediction of CEC ranges from $R^2$ of 0.52 – 0.68. Chang et al. (2001) and Islam et al. (2003) reported $R^2$ of 0.81 and 0.88, respectively for the prediction of CEC. Although some prediction accuracies are slightly lower than other studies, they are still within an acceptable range.

### 3.4. Effect of sample training size: learning curve

A total of nine subset models based on the unique sample sizes were generated to investigate the effect of training sample size.
The performance comparison of all the models expressed as average $R^2$ values is illustrated as a learning curve in Figure 5.
The depicted $R^2$ values are the average performance prediction for all five properties of all ten replicates, except for the largest
sample size (N = 9027) where a single data random split for validation of the data is used. The learning curve generally follows
the common pattern found in machine learning studies (Figueroa et al., 2012), the performance increased rapidly with an
increase in the size of the training set from around 350 to 1400. For PLSR and Cubist, the growth in performance became
slower after it reached 2800 samples. PLSR performance reached a plateau after 4000 samples while the increase in
performance in Cubist was marginal after 5500 samples.
In general, the PLSR and Cubist model tend to perform better when the sample size was relatively small (<1500). When the
sample size was approximately 1800, there was only a small difference in the performances for all models. However, when
the sample size was further increased (>2000), the CNN model started to show better performance in comparison to both PLSR
and Cubist model. The effectiveness of PLSR and Cubist model reached a plateau at approximately 4000 and 5500 samples,
respectively, while the performance of CNN was still increasing, as depicted in the theoretical curve (Figure 1). The slight
drop in Cubist's performance at sample size 9027 was because there was only one realization of data split (75% of the data).
We further compared the average model performance based on the RMSE ratios of machine learning models against the CNN
model (Figure 6). This comparison was developed using the model performance for each unique property, and the variances
presented was based on ten simulations. If a particular model X performs better than the Y model it is compared against, the
RMSE ratios of X/Y should be less than one.
Upon comparing the RMSE ratios of PLSR/Cubist model, we found that PLSR performed better than the Cubist model when
the sample size is less than 1400. Cubist model performed better than the PLSR model as the sample size was increased. Using
the RMSE ratios of PLSR/CNN model, PLSR was found to perform better than CNN when the sample is less than 1400
(Figure 5). Similar performance of both PLSR and CNN model was achieved when the sample size is approximately 1400. In
terms of RMSE ratios of Cubist/CNN, overall CNN model performed better in comparison to the Cubist model regardless of
sample size. This was slightly different than the one that was observed when only $R^2$ parameter was utilized. The RMSE ratios
of Cubist/CNN seemed to vary more for a smaller sample size (longer whisker). When the sample size is approximately 850,
both models seemed to perform similarly. A portion of the model performed better, while the remaining performed worse. As
the calibration sample size increased, the CNN model performed better in comparison to the Cubist model. Thus, it can be
recommended that the current CNN model structure is most efficient for VIS-NIR-SWIR spectral modelling with sample size
above 2000. CNN still can be used for small number of samples, but its performance is not better than PLSR or Cubist.
**3.5.      Sensitivity Analysis**
The critique of CNN is that it is a complex model and a black box. To uncover how the CNN model works, a sensitivity
analysis was conducted to show how CNN is predicting each of the soil properties, illustrated in Figure 7. Only certain parts
of the spectra were used by the CNN model for prediction, which corresponded to the soil properties and composition. The
important wavelengths for the prediction of CEC are between the regions of 1600 – 2000 nm. This result is similar to the
observations made by Lee et al. (2009) on the surface horizon dataset where 1772 and 1805 nm are essential in predicting the
CEC. The presence of high CEC is often linked to the presence of organic matter (OM) and clay content. It is interesting that
the same region is important in predicting organic matter but not clay content.  Aside from the same region used by CEC,
wavelengths' region between 1100 – 1200 nm are also deemed relevant by the CNN model for the prediction of OM content.
This finding is slightly different to those reported by Lee et al. (2009) in which the important wavelengths reported are at 1772,
1871, 2069, 2246, 2351 and 2483 nm for the profile dataset and 1871, 2072 and 2177 nm for the surface horizon dataset.
Similar wavelength regions are deemed to be important in predicting the soil texture although the importance slightly varied
among the type of texture of interest (sand, silt and clay) at wavelengths between 500 and 1800 nm. The important wavelengths
for the prediction of sand and clay content share a higher similarity in comparison to that of silt content prediction. The most
crucial wavelength identified is around 850 nm for the prediction of sand and clay content, and around 1100 nm for the
prediction of silt content. These observations are also different from those reported by Demattê (2002) and Lee et al. (2009)
where the important wavelengths for the prediction of soil texture are at 1800 – 2400 nm. In particular, the soil texture
prediction found in the CNN model is strongly related to hematite and/or goethite, -OH and Al-OH groups from kaolinite
(Viscarra Rossel and Behrens, 2010;Pinheiro et al., 2017;Fang et al., 2018).
We also compare important wavelengths from the machine learning models against the one from the deep learning model for
the prediction of OM as an example. Common wavelengths found to be related to the organic matter predictions are 1100,
1600, 1700 – 1800, 2000, 2200 – 2400 nm (Dalal and Henry, 1986;Stenberg et al., 2010).
As a comparison, we calculated important wavelengths used in the PLSR and Cubist models. The important wavelengths
utilized in the PLSR model was derived based on the absolute value of the regression coefficients. The height of the line
indicates the importance of particular wavelengths for the determination of organic matter content in the soil. Important
wavelengths identified for the prediction of organic matter were 500 – 700, 1400 and 1715 nm.
The wavelengths used in the Cubist were derived based on model usage either as predictors (blue lines) or conditions (pink
lines) (Figure 9). Some of the wavelengths used in the Cubist model are similar to those observed in the PLSR model, in
particular the visible (500 – 700 nm), and shortwave infrared regions (1400 and 1900 nm).
**4.  Discussion**
**4.1.      Understanding the CNN models**
While conventional PLSR and machine learning models require spectra pre-processing for the spectra data input, CNN model
takes raw spectra as inputs. CNN has been shown to be a successful end-to-end learning model which learn feature
automatically while minimizing hand-crafted pre-processing process. Upon taking a closer look at the various filters within
the convolutional layers, we found that the filters behaved like spectra pre-processing method. It is interesting to note that
using the raw spectral input, various spectral pre-processing that was commonly used within spectroscopy could be observed
within the layer itself. Given the various complexity within the CNN model, the use of spectra pre-processing prior to being
fed is unnecessary. This advantage opens up possibilities of developing highly accurate chemometrics model, which also plays
a role in automatic spectral pre-processing.
CNN have been proven to be extremely successful, however how they work remains largely a mystery as they are buried in
layers of computations. Sensitivity analysis enabled us to see better the inner workings of the CNN model. We could understand
better which wavelengths features are essential from the spectra when used in developing the regression prediction. Important
wavelengths derived from the sensitivity analysis based on the CNN model looked slightly different from those of PLSR and
Cubist models. Wavelengths around the 1700 nm region were deemed to be the most important, followed by those between
the 1150 nm region. Nonetheless, some of the important regions overlapped. It was also worth noting that the model did not
use the visible part of the spectra for prediction. In comparison to the sensitivity of MIR spectra data on previous study (Ng et
al., 2019), the NIR model's sensitivity index was much broader, which reflected NIR's characteristic broad peak.
Although all three methods used different ways to derive important wavelengths, PLSR model tended to use most parts of the
spectra. When irrelevant wavelengths are included in model development, it may reduce the model performance. The Cubist
model seemed more selective in terms of wavelengths used, however this example showed that it also used most parts of the
VIS-NIR-SWIR spectra. CNN model used wavelengths between 800 – 2000 nm, with emphasis around 1100 and 1700 nm.

### 4.2. The effect of calibration sample size to model performance

PLSR, Cubist and CNN represent models with increased complexity. By combining results from 5 soil properties, we can
show better a generalisation of the performance of the models as a function of training sample size. Simpler models (PLSR)
performed better at a smaller sample size (< 1400). Cubist outperformed PLSR at sample size > 2000, while CNN outweighed
other models when sample size > 2500. The increase in the accuracy of machine learning models (PLSR and Cubist) became
insignificant when the number of samples was greater than 5000. This trend of plateauing of performance (maximized up to a
certain point) with an increase in sample size as had been observed by several authors (Shepherd and Walsh, 2002;Kuang and
Mouazen, 2012;Ramirez-Lopez et al., 2014;Ng et al., 2018). This trend is related to the complexity of the model, as a simpler
model (such as PLSR) cannot capture all variation in the data. Thus, a more complex model is suitable when the number of
samples is large.
Previous studies by Ng et al. (2019) and Padarian et al. (2019) had shown that CNN performed better than PLSR and Cubist
when the model was trained with more than 10,000 samples. However, there were also studies using CNN with a small number
of training samples. This study showed that CNN model only outperformed PLSR and Cubist models when the sample size is
greater than 2000. As sample size increases, the efficiency of CNN model is increased. We observed a larger reduction in
RMSE (CNN compared to the other 2 models) with increasing calibration sample size. Thus, we recommend using a minimum
of 2000 samples to train CNN model for the VIS-NIR-SWIR spectra. To further improve the performance of the CNN model,
simultaneous prediction of soil properties could also be implemented within the model.
The advantage of using deep learning on a small number of samples is minimal as CNN is a data-hungry model; it is also more
computationally expensive than the typical machine learning models. While our results pertain to the spectral dataset from
Brazil and a particular structure of the CNN, we believe our results can serve as a guide on the number of samples needed to
create a better deep learning model. Future research could test this idea on larger and more variable datasets (e.g. a global
spectral library with more than 100,000 samples) and to see if a more complex and deeper network of CNN can handle such
dataset.

## 5. Conclusions

In this paper, we assessed the effect of training sample size and identified important wavelengths in predicting various soil
properties using Cubist and CNN model. In general, the CNN model performed better than the Cubist when the sample size is
relatively large (>2000). Here, we found that with its current model structure, CNN is more accurate than a machine learning
model when the number of calibration samples is above 2000. The more complex and deeper network of a deep learning model,
the more likely it will need a larger number of samples for training. PLSR and Cubist models perform less accurate than the
CNN model as sample size increases, and both models reached a plateau after a sample size of 4000 – 5000. Meanwhile, the
performance of CNN still increases until the maximum number of data used in this study (N = 9000). Future studies should
explore larger dataset to see the generalization of the accuracy vs sample size and to explore if the deep learning CNN model
ever reaches a plateau in accuracy.

## Author contributions

Wartini Ng was responsible for the data analysis, and prepared the manuscript; Budiman Minasny contributed to the idea, data
analysis and editing the manuscript; Wanderson de Sousa Mendes and José A.M.Demattê contributed to the idea, provided the
data and editing the manuscript.
**Competing interests**
The authors declare that they have no conflict of interest.
**Acknowledgements**
This study was financed in part by the ARC Linkage Project LP150100566 - Optimised field delineation of contaminated soils.
The authors would also like to thank members of the Geotechnologies in Soil Science Group
(https://esalqgeocis.wixsite.com/geocis) and Sao Paulo Research Foundation (FAPESP, grant numbers 2014/22262-0 and
2016/26124-6). BM is a member of a consortium supported by LE STUDIUM Loire Valley Institute for Advanced Studies
through its LE STUDIUM Research Consortium Programme.

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

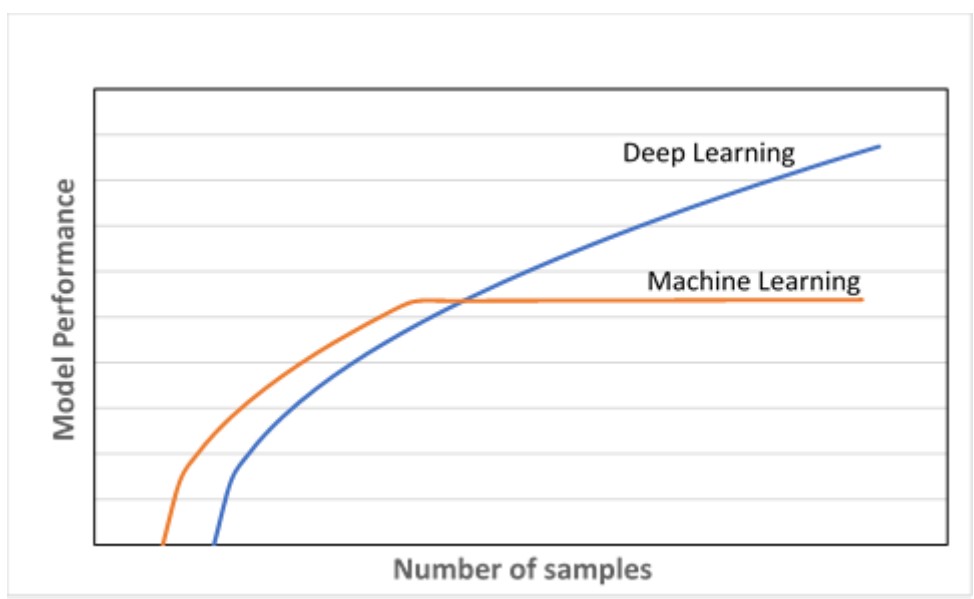


**Figure 1. Model performance of deep learning vs other machine learning algorithms as a function of number of samples.**

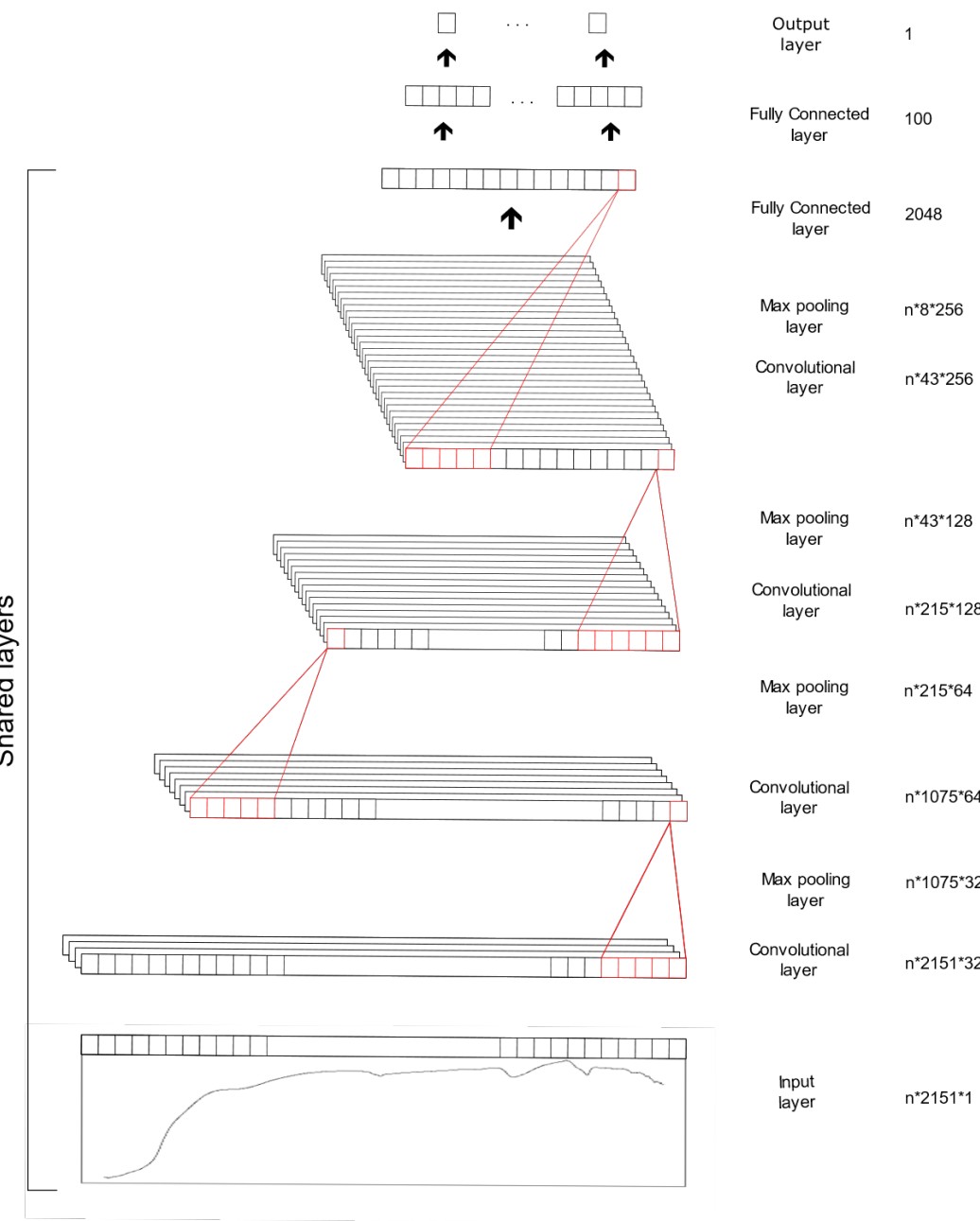


**Figure 2. Architecture of the one-dimensional Convolutional Neural Network (CNN) model.**


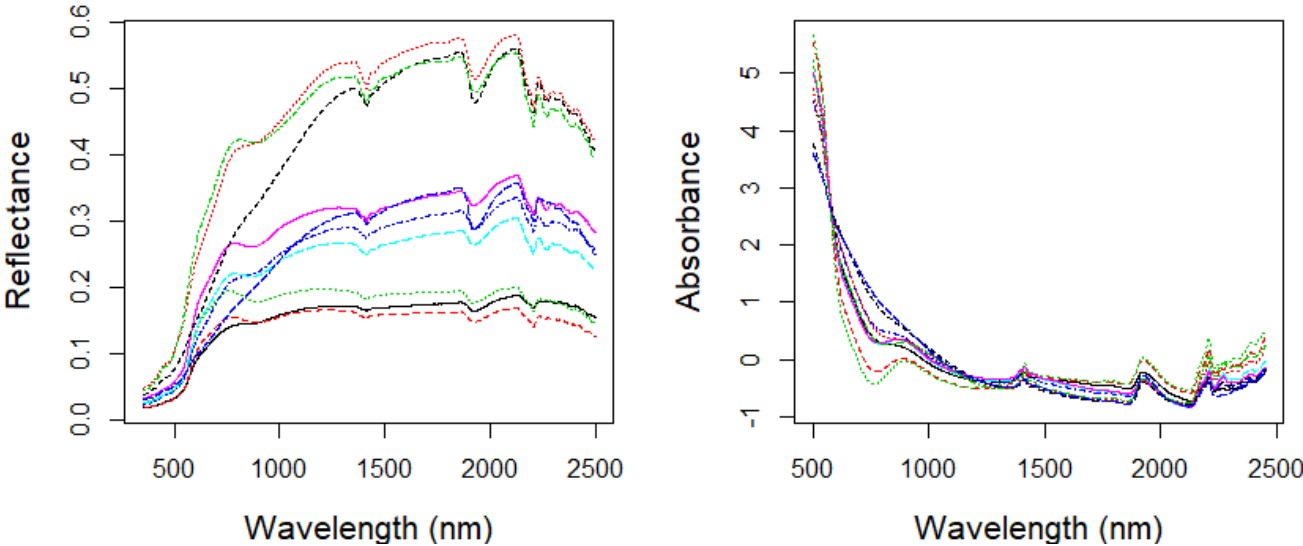

**Figure 3. Visible, near and shortwave infrared (VIS-NIR-SWIR) spectra of 10 soil samples without spectra pre-processing (left) and**
**with spectra pre-processing (right).**



Convolution #1: A few of the 32 filters

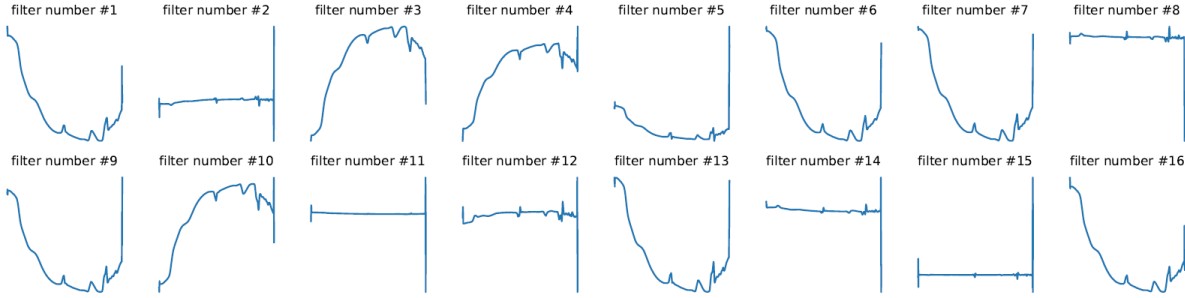


Convolution #2: A few of the 64 filters

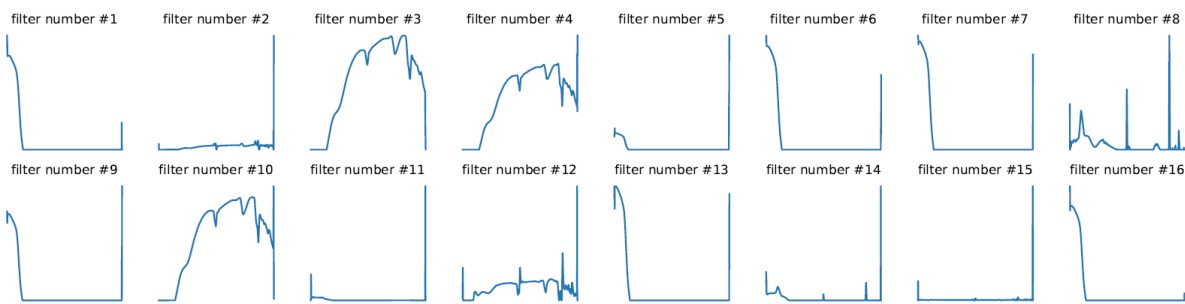


**Figure 4. Visualization of the filters in the first two convolutional layers within the one-dimensional Convolutional Neural Network**
**(CNN) model of the visible, near, and shortwave infrared (VIS-NIR-SWIR) spectra data.**

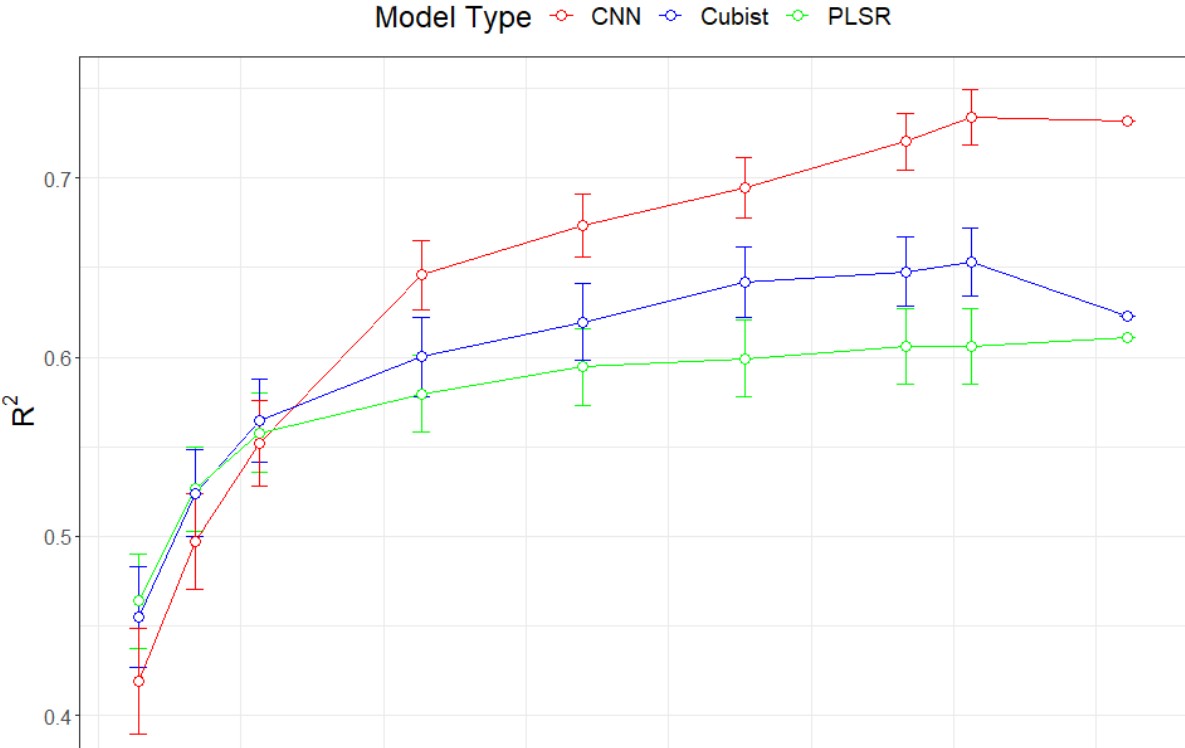


**Figure 5. Model performances (in terms of average $R^2$ for five soil properties) as a function of sample size using Partial Least Squares**
**Regression (PLSR), Cubist and Convolutional Neural Network (CNN) model based on ten simulations. The value for the largest**
**sample size (N = 9027) is a single realization 75% of the data.**

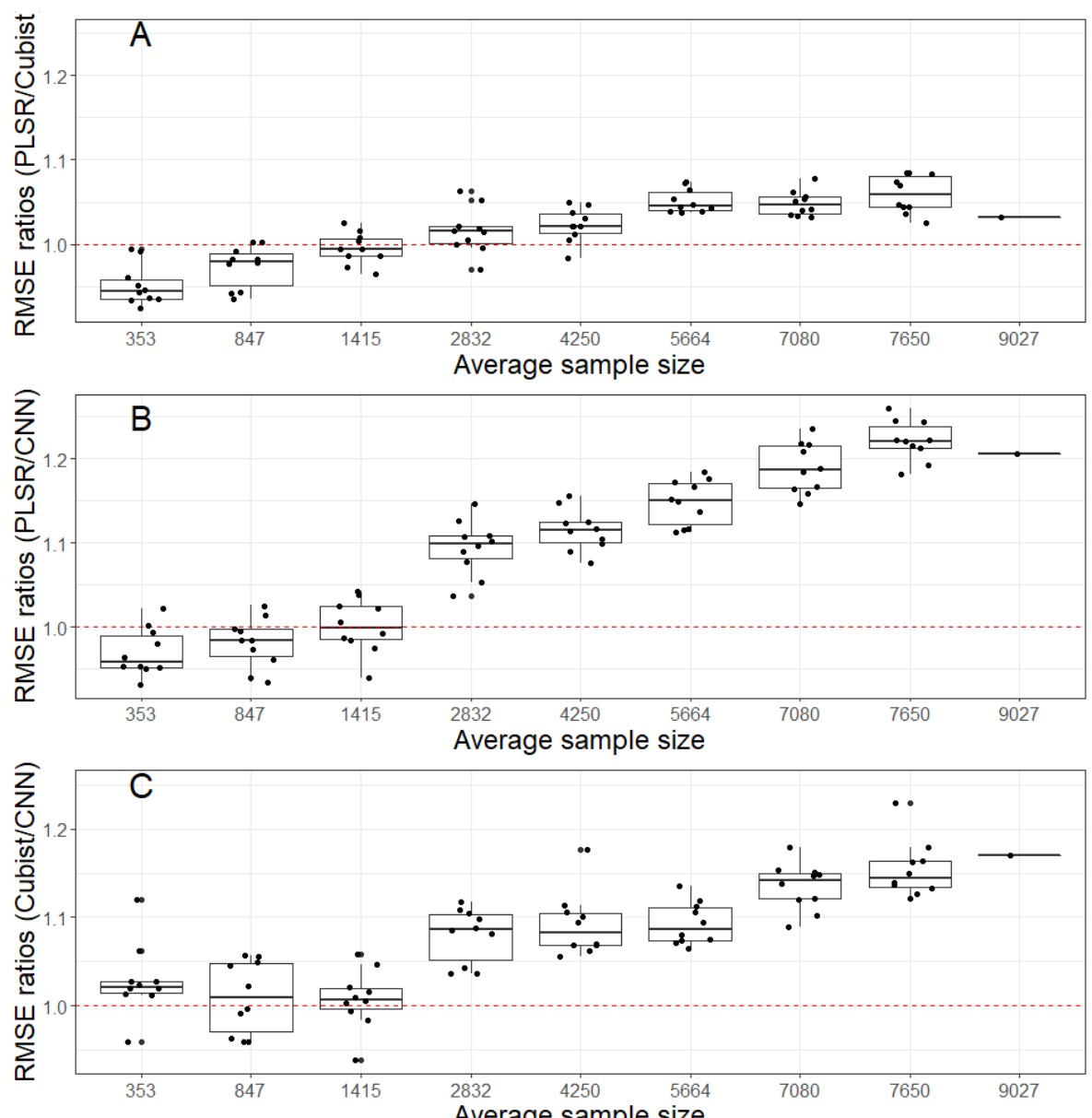


**Figure 6. Model performances (in terms of root mean square error (RMSE) ratios of (A) Partial Least Squares Regression (PLSR)**
**over Cubist model (B) PLSR over Convolutional Neural Network (CNN) model and (C) Cubist over CNN as an average of five soil**
**properties) based on various sample size using ten simulations. The red – dotted line represents a 1:1 RMSE ratio.**

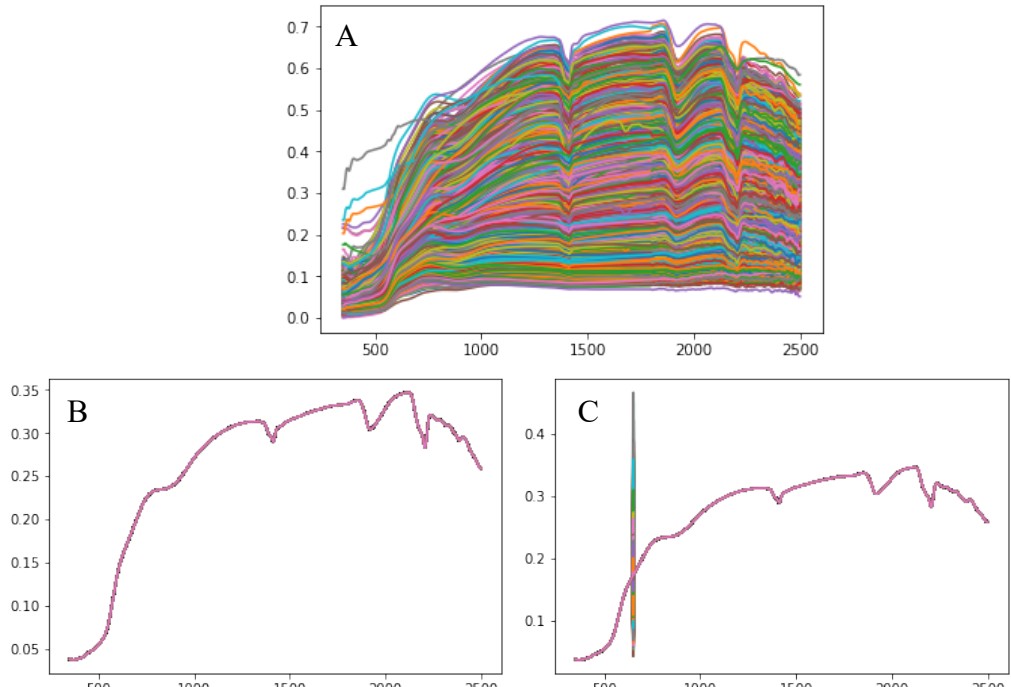

**Figure 7. Illustration of sensitivity analysis process: (A) represents the validation spectra data, (B) represents the overall average of**
**the validation spectra data and (C) represents the modified average of the validation spectra data.**

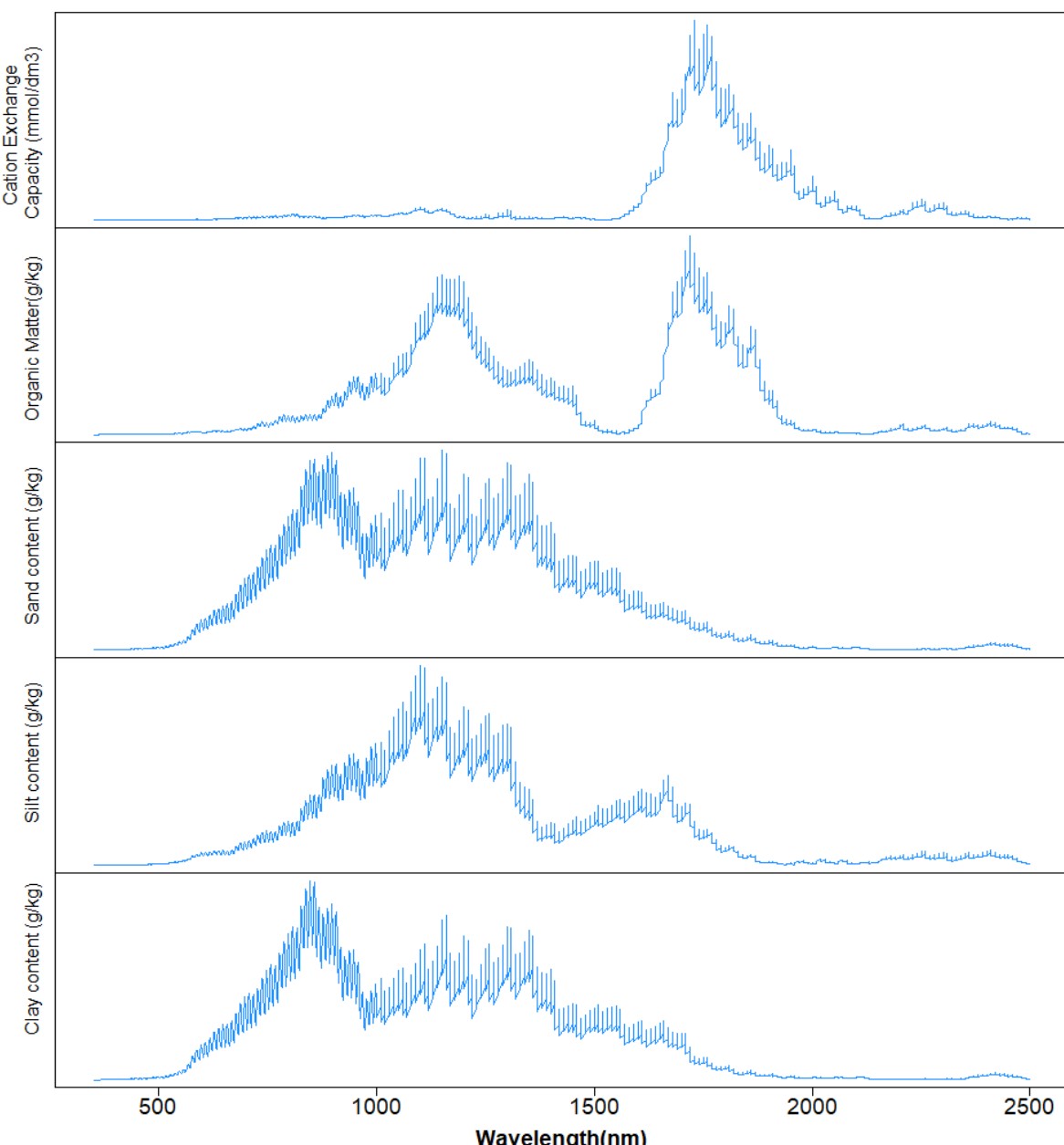


**Figure 8: Sensitivity analysis of the visible, near and shortwave infrared (VIS-NIR-SWIR) spectra in predicting various soil properties using the Convolutional Neural Network (CNN) model. The graph depicts sensitivity index (calculated from(Eq. 2)) for different soil properties as a function of wavelength.**


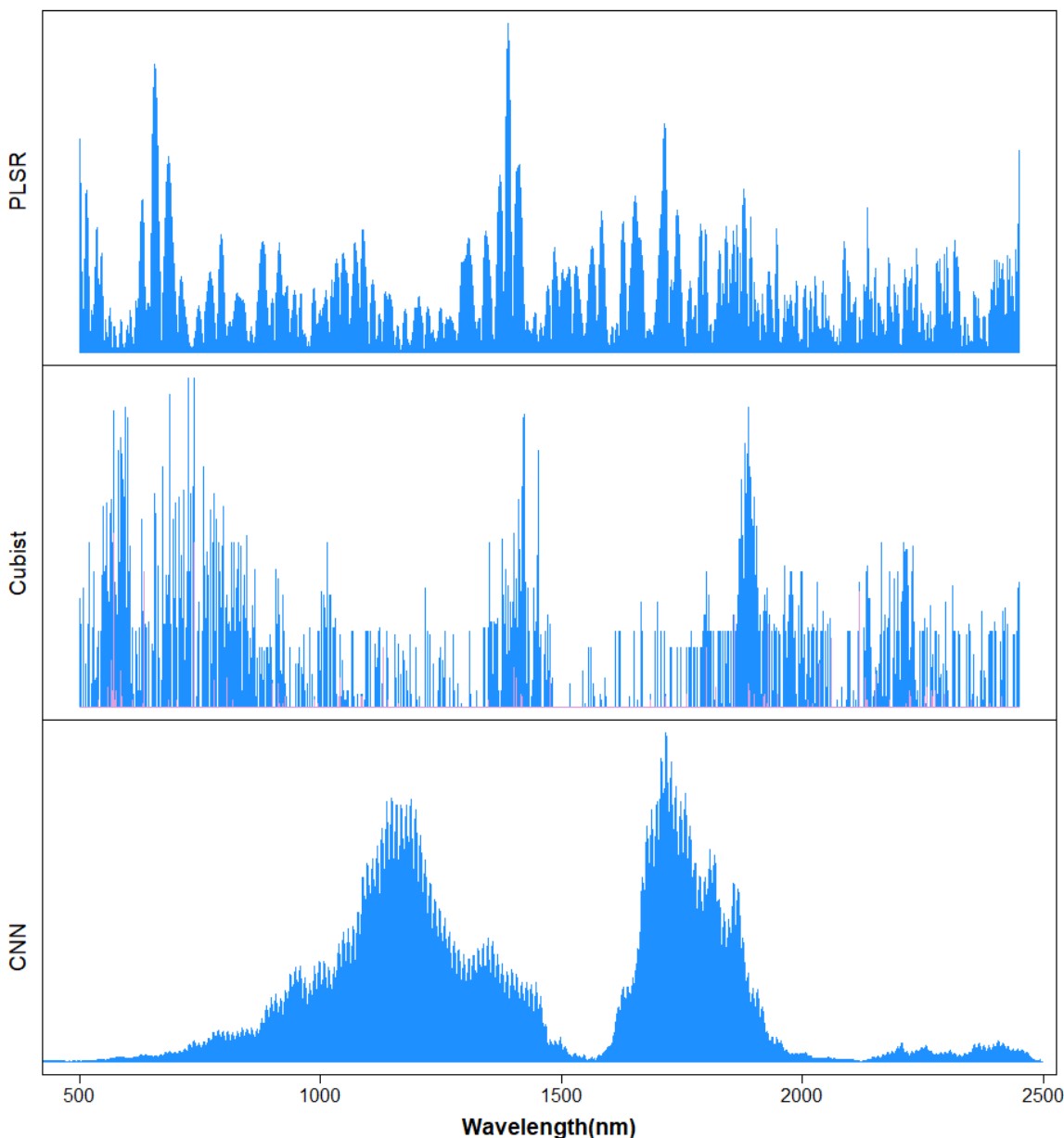


**Figure 9: Important wavelengths for the prediction of organic matter (OM) content using Partial Least Squares Regression (PLSR),**

**Cubist and Convolutional Neural Network (CNN) model.**




**Table 1: Descriptive statistics of the soil properties measurements.**

| | Sand | Silt | Clay | OM | CEC |
|---|---|---|---|---|---|
| | g kg$^{-1}$ | | | mmol$_c$ kg$^{-1}$ | |
| **Minimum** | 50.0 | 0.0 | 5.0 | 2.0 | 3.4 |
| **1$^{st}$ Quartile** | 644.0 | 31.0 | 112.0 | 6.0 | 22.9 |
| **Median** | 757.0 | 57.0 | 174.7 | 9.4 | 32.7 |
| **Mean** | 703.8 | 69.7 | 226.5 | 11.2 | 37.7 |
| **3$^{rd}$ Quartile** | 839.0 | 93.5 | 283.3 | 14.3 | 46.3 |
| **Maximum** | 969.0 | 562.0 | 840.0 | 69.0 | 375.7 |



**Table 2: Architecture of the convolutional neural network.**

| Type | Shared | Filter size | # Filters | Activation |
|---|---|---|---|---|
| Convolutional | Yes | 20 | 32 | ReLU |
| Max-pooling | Yes | 2 | - | - |
| Convolutional | Yes | 20 | 64 | ReLU |
| Max-pooling | Yes | 5 | - | - |
| Convolutional | Yes | 20 | 128 | ReLU |
| Max-pooling | Yes | 5 | - | - |
| Convolutional | Yes | 20 | 256 | ReLU |
| Max-pooling | Yes | 5 | - | - |
| Dropout (0.4) | Yes | - | - | - |
| Flatten | Yes | - | - | - |
| Fully-connected | No | - | 100 | ReLU |
| Dropout (0.2) | No | - | - | - |
| Fully-connected | No | - | 1 | Linear |

*ReLU: rectified linear units



**Table 3: Results of model validation for the prediction of various soil attributes using the full calibration dataset.**

| Model | Properties | Unit | $R^2$ | RMSE | bias | RPIQ |
|-------|-----------|------|-------|------|------|------|
| PLSR | Sand | | 0.79 | 91.47 | 2.74 | 1.29 |
| | Silt | g kg$^{-1}$ | 0.47 | 41.58 | -1.78 | 0.67 |
| | Clay | | 0.80 | 73.01 | -0.65 | 0.87 |
| | OM | | 0.48 | 4.98 | 0.04 | 0.70 |
| | CEC | mmol$_c$ kg$^{-1}$ | 0.52 | 16.77 | -0.17 | 0.57 |
| Cubist | Sand | | 0.78 | 89.66 | 1.28 | 1.19 |
| | Silt | g kg$^{-1}$ | 0.45 | 38.68 | -2.06 | 0.67 |
| | Clay | | 0.81 | 69.65 | -0.23 | 0.92 |
| | OM | | 0.54 | 4.83 | -0.22 | 0.70 |
| | CEC | mmol$_c$ kg$^{-1}$ | 0.52 | 17.03 | -0.93 | 0.59 |
| CNN | Sand | | 0.85 | 77.28 | -0.16 | 1.52 |
| | Silt | g kg$^{-1}$ | 0.58 | 37.09 | -1.74 | 0.75 |
| | Clay | | 0.86 | 60.78 | -0.53 | 1.05 |
| | OM | | 0.69 | 3.83 | -0.11 | 0.91 |
| | CEC | mmol$_c$ kg$^{-1}$ | 0.68 | 13.73 | -0.76 | 0.69 |

OM = organic matter; CEC = cation exchange capacity

