# Peer review of "The influence of training sample size on the accuracy of deep learning"

_SOIL, 2019_

## Referee Comment (RC1) · Anonymous Referee #1 · 19 Oct 2019

The paper presents a case study of prediction performance comparisons between a couple of standard machine learning methods commonly used in soil spectroscopy (PLS and Cubist) and a deep learning algorithm (convolutional neural networks, CNN). These algorithms are tested in a large soil spectral library. The paper clearly shows that CNN outperforms PLS and Cubist when the number of calibration observations is large. All the algorithms tend to perform poorly when they are used in small (calibration) sample sets. In my opinion, the manuscript does not present a clear contribution to soil science. I just have some comments that I hope help the authors to improve their manuscript.

General comments:

- The method used by the authors to estimate the effective calibration sample size is entirely based on prediction performance indicators (e.g. root mean square error) which requires prior knowledge of the response variables of the samples used as candidates for calibration. Therefore, the method is rather unrealistic/impractical.

- Since the main objective of the paper is related to calibration sampling for soil spectroscopy, I encourage the authors to review the literature available on this topic. This might help to clearly identify research needs and also to identify already available methods to optimize the amount of samples used in calibration (see Esbensen et al., 2014; Ramirez-Lopez et al., 2014; De Gruijter et al., 2006 ; Petersen et al., 2005; Minkkinen, 2004).

- The effective size of the calibration set for a given spectral dataset largely depends on the variability or complexity embedded in such dataset. For example, a small area where a large number of soil spectra is available (as in the case of on-the-go soil spectroscopy), the optimal size of the calibration set would be rather small. Furthermore, in such non-complex scenario, the use of CNN would be arguable, as the conventional methods would be expected to perform well (as it has been proven). In this respect, the authors seem to focus only on the size of the calibration sets disregarding very important aspects of the theory of sampling (see Minkkinen, 2004) and draw general conclusions from a single experimental dataset.

- The conclusions are not clear and despite their original research question (how many samples are required to get CNN performing better than PLS and Cubist) is answered for their particular dataset, there is no useful procedure or method presented by the authors to reproduce or extrapolate this to other cases in a useful way.

Specific comments:

- Section 3 (chemometrics model): the authors need to provide information on model

optimization and references. For example, why do they choose a learning rate of 0.001 and adam optmizer, what does it mean? Is there any reference the readers can be referred to? How many PLS components and committees were tested in PLS and cubist respectively? Optimization of the algorithms play a key role in their performance.

- Section 4.4 (sensitivity analysis): this whole section does not seem to bring any significant contribution to the objectives of the paper.

- Section 4.4 (sensitivity analysis): The estimations of the importance of variables for modeling for different modeling algorithms are based on different methods, therefore the comparisons between the results carried out in the paper are not appropriate.

- Section 4.4 (sensitivity analysis): the authors need to be more clear with their statement: "the wavelengths used Cubist were derived based on model usage".

References

De Gruijter, J., Brus, D. J., Bierkens, M. F., & Knotters, M. (2006). Sampling for natural resource monitoring. Springer Science & Business Media.

Esbensen, K. H., & Wagner, C. (2014). Theory of sampling (TOS) versus measurement uncertainty (MU)–A call for integration. TrAC Trends in Analytical Chemistry, 57, 93-106.

Minkkinen, P. (2004). Practical applications of sampling theory. Chemometrics and intelligent laboratory systems, 74(1), 85-94.

Ng, W., Minasny, B., Montazerolghaem, M., Padarian, J., Ferguson, R., Bailey, S., & McBratney, A. B. (2019). Convolutional neural network for simultaneous prediction of several soil properties using visible/near-infrared, mid-infrared, and their combined spectra. Geoderma, 352, 251-267.

Padarian, J., Minasny, B., & McBratney, A. B. (2019). Using deep learning to predict soil properties from regional spectral data. Geoderma Regional, 16, e00198.

Petersen, L., Minkkinen, P., & Esbensen, K. H. (2005). Representative sampling for reliable data analysis: theory of sampling. Chemometrics and intelligent laboratory systems, 77(1-2), 261-277.

Ramirez-Lopez, L., Schmidt, K., Behrens, T., Van Wesemael, B., Demattê, J. A., & Scholten, T. (2014). Sampling optimal calibration sets in soil infrared spectroscopy. Geoderma, 226, 140-150.

---

## Referee Comment (RC2) · Anonymous Referee #1 · 19 Oct 2019

Two recently published papers (by some of the authors of the present paper, see Padarian et, al. 2019 and Ng et al., 2019) have shown already that for soil spectroscopy, the CNN algorithm tend to outperform PLS and Cubist algorithms for modeling large soil spectral libraries.

---

## Referee Comment (RC3) · Anonymous Referee #2 · 16 Mar 2020

The manuscript tackles with an important and interesting topic; however, the presentation was really poor, not easy to follow. The most important issue is that the manuscript lacks the Discussion section! Actually the manuscript is not ready to be submitted to a journal.

- Apart from the Abstract and Introduction sections, the other sections were totally mixed in a way that in some parts you could not get, which section you are reading. For example, Lns. 176-196 are method but have been presented in the Results sections. This is a critical issue in a paper that needs to be solved.

- The authors have compared CNN with PLSR and Cubist, as two common machine

learning techniques, although Cubist have not been very common in soil spectroscopy so far compared to RF and SVM. It would be fine if these algorithms also be taken into account.

- Some parts repeating the same thing several times. For instance, the section 4.3. generally repeats the same contents in Lns. 158-163 and Lns. 168-173 that should be avoided.

- In presenting the comparison between PLSR and Cubist has been missed. Please compare them as well. In general, the Results sections should be more detailed furnished with more obtained values and comparison of them.

- Surprisingly, the manuscript does not have the Discussion section, which is one of the most important parts of each paper. There are only some lines in the Result section whitin authors have presented the results of other similar studies (e.g. Lns. 148-151, Lns. 198-207, Lns. 212-215), which cannot be considered as the discussion of the results of the current work. Please separate the section of Results from the Discussion with detailed and informative discussion of your works' outputs.

All to all, I reject the manuscript at this step but highly recommend its resubmission after the corrections done.

---

## Author Comment (AC3) · 8 Apr 2020

Two recently published papers (by some of the authors of the present paper, see Padarian et, al. 2019 and Ng et al., 2019) have shown already that for soil spectroscopy, the CNN algorithm tend to outperform PLS and Cubist algorithms for modeling large soil spectral libraries.

We agreed that two of the recently published papers have shown that CNN outperform both PLS and Cubist in the large data set. The objective of this paper is to show that CNN has lower performance when sample size is small, and estimate the effective sample size on which CNN would out perform the other two models.

---

## Author Response (AR1)

Thank you for taking the time to review our manuscript. We will address the comments and revise the paper accordingly. Our detailed responses are as follow:

**The paper presents a case study of prediction performance comparisons between a couple of standard machine learning methods commonly used in soil spectroscopy (PLS and Cubist) and a deep learning algorithm (convolutional neural networks, CNN). These algorithms are tested in a large soil spectral library. The paper clearly shows that CNN outperforms PLS and Cubist when the number of calibration observations is large. All the algorithms tend to perform poorly when they are used in small (calibration) sample sets. In my opinion, the manuscript does not present a clear contribution to soil science. I just have some comments that I hope help the authors to improve their manuscript.**

We agree that the comparison on the performance of deep learning vs machine learning (Cubist and PLS) using a large NIR spectra library to predict soil properties has been published by our group (Padarian et al and Ng et al., 2019). Deep learning application in soil spectroscopy, and even in spectroscopy is still new (Yang et al. 2019)

Currently, there is still no guideline on how many samples would we need to effectively use deep learning methods. Deep learning was developed to handle a large amount of data (millions of images), and clearly soil spectra data are not that large. For example, a recent study used deep learning on 135 soil samples (Chen et al., 2018). Clearly the advantage of using deep learning on such small number of samples is questionable. A recent review on spectroscopy showed that there are a large number of studies where deep learning was used with small sample size (Yang et al. 2019). The review indicated that increased training samples could further improve the calibration performance, however there is no guideline how much improvement can be expected and what is the minimum number of samples.

In addition, there is a hypothesis in machine learning literature that common regression methods will reach a plateau with increasing sample size, while the performance of deep learning will still increase. We tested this hypothesis in soil NIR modelling,

Hence, the contribution to soil science is:

- Establishing the number of samples required for deep learning to be effective
- To test the hypothesis that common machine learning models with reach a plateau in accuracy with an increasing number of samples.
- Establishing how much improvement in accuracy when we increase the number of calibration sample
- Demonstrating how to interpret deep learning models

Chen, H., Liu, Z., Gu, J., Ai, W., Wen, J. and Cai, K., 2018. Quantitative analysis of soil nutrition based on FT-NIR spectroscopy integrated with BP neural deep learning. Analytical methods, 10(41), pp.5004-5013.

Yang, Jie, Jinfan Xu, Xiaolei Zhang, Chiyu Wu, Tao Lin, and Yibin Ying. "Deep learning for vibrational spectral analysis: Recent progress and a practical guide." Analytica Chimica Acta (2019).

**General comments: - The method used by the authors to estimate the effective calibration sample size is entirely based on prediction performance indicators (e.g. root mean square error) which requires prior knowledge of the response variables of the samples used as candidates for calibration. Therefore, the method is rather unrealistic/impractical.**

The objective of this paper is not on calibration sampling or estimating the effective number of sample size for an area as done by Ramirez-Lopez et al., 2014. We recognise the title would need a revision, which we will revise.

The aim of the application of spectroscopy in soil science field is to provide rapid prediction of soil properties. In order to do it, prior knowledge of the response variables is indeed needed. There is no other way of estimating the effective calibration size rather than using empirical data.

**- Since the main objective of the paper is related to calibration sampling for soil spectroscopy, I encourage the authors to review the literature available on this topic. This might help to clearly identify research needs and also to identify already available methods to optimize the number of samples used in calibration (see Esbensen et al., 2014; Ramirez-Lopez et al., 2014; De Gruijter et al., 2006 ; Petersen et al., 2005; Minkkinen, 2004).**

The objective of this paper is not on calibration sampling or estimating the effective number of sample size for an area. This paper aims to provide a guide on the number of samples to be used within the calibration model for both deep learning and machine learning model in a large and diverse dataset. In our previous study, we have found that the machine learning model reaches plateau performance when the number of sample size reach several thousand. We would like to understand a comparison of performance of machine learning vs. deep learning with number of samples.

**- The effective size of the calibration set for a given spectral dataset largely depends on the variability or complexity embedded in such dataset. For example, a small area where a large number of soil spectra is available (as in the case of on-the-go soil spectroscopy), the optimal size of the calibration set would be rather small. Furthermore, in such non-complex scenario, the use of CNN would be arguable, as the conventional methods would be expected to perform well (as it has been proven). In this respect, the authors seem to focus only on the size of the calibration sets disregarding very important aspects of the theory of sampling (see Minkkinen, 2004) and draw general conclusions from a single experimental dataset.**

We agreed that for a smaller area, the use of conventional machine learning would be suitable. We did not disregard the theory of the sampling and as described above this paper is not about establishing sampling for calibration. The sample selected for the calibration model is based on stratified sampling scheme, where the samples from the same sites are grouped together.

**- The conclusions are not clear and despite their original research question (how many samples are required to get CNN performing better than PLS and Cubist) is answered for their particular dataset, there is no useful procedure or method presented by the authors to reproduce or extrapolate this to other cases in a useful way.**

Although the conclusion we drew applied only for this dataset, it would provide the readers a guide of when to and not to use the CNN model within their own dataset. Clearly CNN requires a large dataset. It would not be possible to analyse all combination of spectra library. We believe the 2000 samples are applicable as a general guide.

**Specific comments: - Section 3 (chemometrics model): the authors need to provide information on model optimization and references. For example, why do they choose a learning rate of 0.001 and adam optmizer, what does it mean? Is there any reference the readers can be referred to?**

We will provide this information in the revised paper. We need to ensure that the learning rate is not too high or too low. Adam is one of the learning optimizer that is used when training neural network (aside from RMSprop, SGD, Adadelta, etc.) For more information regarding this optimizer, refer to Kingma and Ba (2014).

Kingma, D. P., and Ba, J.: Adam: A method for stochastic optimization, arXiv preprint arXiv:1412.6980, 2014.

**How many PLS components and committees were tested in PLS and cubist respectively? Optimization of the algorithms play a key role in their performance.**

We could not agree more that the optimization is important. We had included this within the paper. For the PLS model, we selected number of components that resulted in the lowest RMSE based on cross validation approach. The committees for the Cubist model was set as 1.

**- Section 4.4 (sensitivity analysis): this whole section does not seem to bring any significant contribution to the objectives of the paper.**

We still think this is an important part of modelling. This section is not related to sampling size for both machine learning and deep learning model, however interpretation is as much important as getting a highly accurate prediction. We would like to demonstrate that deep learning does not necessarily be a black box. We show that the sensitivity analysis can relate the model back to the basic knowledge. The sensitivity of certain region can be related to the presence of certain molecules that affect the prediction of certain soil properties.

**- Section 4.4 (sensitivity analysis): The estimations of the importance of variables for modeling for different modeling algorithms are based on different methods, therefore the comparisons between the results carried out in the paper are not appropriate.**

We couldn't agree more. We recognise the differences and that each method is unique, and it is mentioned in the paper. For example, PLS regression parameters method cannot be applied to Cubist and CNN. We just want to mention that there are different methods to interpret the model.

**- Section 4.4 (sensitivity analysis): the authors need to be more clear with their statement: "the wavelengths used Cubist were derived based on model usage".**

We have clarified it in our revised paper as: The important wavelengths were selected based on the variables that were used within the model either as predictors (blue lines) or conditions (pink lines).
Thank you for taking the time to review our manuscript. We will address the comments and revise the paper accordingly. Kindly find our detailed responses as follow:

**The manuscript tackles with an important and interesting topic; however, the presentation was really poor, not easy to follow. The most important issue is that the manuscript lacks the Discussion section! Actually the manuscript is not ready to be submitted to a journal.**

We will elaborate the discussion section.

**- Apart from the Abstract and Introduction sections, the other sections were totally mixed in a way that in some parts you could not get, which section you are reading. For example, Lns. 176-196 are method but have been presented in the Results sections. This is a critical issue in a paper that needs to be solved.**

We will move it into Method section, and keep the results within Result section.

**- The authors have compared CNN with PLSR and Cubist, as two common machine learning techniques, although Cubist have not been very common in soil spectroscopy so far compared to RF and SVM. It would be fine if these algorithms also be taken into account.**

We'll remove the word "common method" when referring to Cubist model. Several of the studies below had shown that Cubist > RF and SVM. Hence, the inclusion of RF and SVM would not be necessary.

> Sorenson, P. T., Small, C., Tappert, M. C., Quideau, S. A., Drozdowski, B., Underwood, A., and Janz, A. 2017. Monitoring organic carbon, total nitrogen, and pH for reclaimed soils using field reflectance spectroscopy. Canadian Journal of Soil Science, 97(2), 241-248.

> Sharififar, A., Singh, K., Jones, E., Ginting, F. I., and Minasny, B. 2019. Evaluating a low-cost portable NIR spectrometer for the prediction of soil organic and total carbon using different calibration models. Soil Use and Management, 35(4), 607-616.

> Silva, E. B., Giasson, É., Dotto, A. C., Caten, A. t., Demattê, J. A. M., Bacic, I. L. Z., and Veiga, M. d. 2019. A Regional Legacy Soil Dataset for Prediction of Sand and Clay Content with Vis-Nir-Swir, in Southern Brazil. Revista Brasileira de Ciência do Solo, 43.

**- Some parts repeating the same thing several times. For instance, the section 4.3. generally repeats the same contents in Lns. 158-163 and Lns. 168-173 that should be avoided.**

We understand that it seems that we are repeating the same thing several times. However, $R^2$ itself is not enough in chemometrics. Thus, RMSE are also implemented to evaluate model performance.

In our example, although it seems that Cubist and PLSR performed better than the CNN model in terms of $R^2$ for smaller sample size (see Figure 5); there are larger variances in the RMSE of Cubist model in comparison to CNN model. Based on the $R^2$ itself, both PLSR and Cubist seemed to also perform similar. However, when we compare the model performance in terms of RMSE ratios, we can see that there are less variances using the PLSR model.

**- In presenting the comparison between PLSR and Cubist has been missed. Please compare them as well. In general, the Results sections should be more detailed furnished with more obtained values and comparison of them.**

Yes, we shall include the comparison between the PLSR and Cubist model within the script as well ( see updated figure 6)

**- Surprisingly, the manuscript does not have the Discussion section, which is one of the most important parts of each paper. There are only some lines in the Result section whitin authors have presented the results of other similar studies (e.g. Lns. 148-151, Lns. 198-207, Lns. 212-215), which cannot be considered as the discussion of the results of the current work. Please separate the section of Results from the Discussion with detailed and informative discussion of your works' outputs.**

We'll elaborate the discussion as we have mentioned above..

**All to all, I reject the manuscript at this step but highly recommend its resubmission after the corrections done.**

We have revised the paper, and added discussion of our findings.

[revised manuscript text omitted]

Thus, the purpose of this study is to assess the amount of calibration data needed for the CNN model to perform better than machine learning models. PLSR and Cubist are chosen as the representatives of the regression and machine learning models which has been commonly used to develop predictive models based on soil spectra data.machine learning models which had been found to perform well in soil spectra data (e.g., Dangal et al. (2019)). In addition, to be able to predict soil properties accurately, we need to understand and interpret how a CNN model can predict soil properties from spectra. The sensitivity analysis of the VIS-NIR-SWIR region used in the CNN model is performed to uncover the CNN black box.
[revised manuscript text omitted]
 is conducted. The reflectance spectrum data was fed into the first convolutional layer. The filter in the first layer encodes various pre-processing of the input spectra data. Some of the filters shown in the first convolution layer looks like the input spectra pattern (filter #3, 4 and 10), and some of them looks like transformation pattern: absorbance (filter #1, 5, 6, 7, 9, 13 and 16) and derivatives (filter # 2, 8, 11, 12, 14 and 15). The spectrum becomes smoother when they passed through the second convolutional layer, where some filters only accentuate certain peaks (Figure 3). Thus, the ability of the convolutional layers to represent various transformation of the spectra make CNN a robust model that does not require any spectra pre-processing.

**Model performance comparison**

The model performances for the validation dataset using the full calibration data ($n_{site}$= 3188, N=9027) with all the models are first presented in Table 3. Among all the properties predicted, the sand and clay content showed the best performance with $R^2$ values greater than 0.75 regardless of the types of model used. This finding is in agreement with the ones from Demattê et al. (2016), who observed good predictions for sand and clay content.

Demattê et al. (2016) reported $R^2$ values ranging from 0.51 and 0.86 for sand (0.86), silt (0.51, clay (0.85), organic matter (0.63) and CEC (0.66) using PLSR model with 4790 out of 7185 samples as calibration samples. The performances of our PLSR and Cubist model are lower than those reported by Demattê et al. (2016) could probably due to the larger variation of the dataset used here. Furthermore, representative sampling using conditioned Latin hypercube sampling was used in selecting the calibration samples prior to the model development. Nonetheless, the overall CNN model used here still performs better.

**Effect of sample training size: sub-setting the calibration data**

A total of eight subset models based on the unique sample sizes were generated. The performance comparison of CNN and Cubist model based on average $R^2$ values is illustrated in Figure 4. The reported $
[revised manuscript text omitted]

OM = organic matter; CEC = cation exchange capacity

---

## Author Response (AR2)

The paper discusses the performance of convolutional neural networks (CNN) compared to traditional machine learning techniques in function of the number of calibration samples. The first reviewer of the previous version misunderstood the objectives and therefore had serious doubts on the novelty and the strategies. The authors have changed the title and clarified the objectives in order to deal withthis misunderstanding.

Thank you for taking the time to review our manuscript. We shall address the comments and revise the paper accordingly. Our detailed responses are as follow:

The second reviewer mentions a lack of a discussion section. This has now been added.

Specific comments

Line 59, 85-86, 130, 139 …. (please check throughout the anuscript) …spectral data… or 'spectra' on its own.

Response: We have checked the consistency through-out the text.

Line 59-60 …predict clay content..

Response: We have corrected this.

Line 64 …soil spectral libraries…

Response: We have corrected this.

Line 68 What are 'increased calibration samples'? Is there a word missing?

Response: We have corrected this. We are refererring to calibration sample size.

Line 68 Split up the sentence: …performance. However, there ….

Response: We have split the sentence.

Line 70 A strategy …

Response: We have added the article "A" as suggested.

Line 73 …. How many samples….

Response: We have corrected the word 'much' to 'many'

Line 75 CNN will outperform traditional…

Response: We have corrected this

Line 78 …CNN model to outperform machine….

Response: We have corrected this

Lines 81, 82 Delete either 'specifically' or 'specific'

Response: We have removed one of the word

Line 83 ..models will reach..

Response: We have corrected this

Line 87 There seems to be a word missing : …achieved wen the number…

Response: We have added a word

Line 92 Delete 'of'

Response: We have deleted the word.

Line 95 …and sedimentary rocks..

Response: We have corrected the word

Line 95 Delete 'samples'

Response: We have deleted this

Line 102 In general it is better to work as much as possible with the original data. There is no need to multiply all organic carbon contents with 1.724. I would prefer if you use soil organic carbon (SOC) for all prediction models. After the Van Bemmelen factor is an empirical one and adds unnecessary noise to the data.

Response: We reported the value in OC as suggested

Line 103 .. to extract exchangeable aluminium, calcium and magnesium…

Response: We have corrected this

Line 117 …spectral measurement…

Response: We used the word spectra through-out

Line 152 …as one-dimensional data…

Response: We have corrected this

Line 156 Word missing : …was trained using a batch…

Response: We have added the word

Line 200 ….each filter of…

Response: We have corrected this

Line 260 I do not understand what you want to illustrate in Fig. 7. The sensitivity analysis is shown in Fig.8.

Response: We have corrected the reference for the figure

Line 333 Delete 'in this paper'

Response: We have removed the word.

Anonymous Referee #1

Thank you for taking the time to review our manuscript. Kindly find our responses below:

- I still think comparison with RF and SVM needs to be conducted as many studies have shown the superiority of these algorithms rather than other machine learing techniques. Moreover, as many researchers in soil community use these machine learning techniques, they need to know their performance compared to deep learning.

Response:

The aim of this paper is not to compare various machine learning algorithms. There are many papers who have done that already. The aim of this paper is to assess the effect of training sample size on the accuracy of deep learning and compare it with some machine learning models as benchmark.

Here we included findings from other researchers that conducted the study as requested by the reviewer for various soil properties below.

Silva, E. B., Giasson, É., Dotto, A. C., Caten, A. t., Demattê, J. A. M., Bacic, I. L. Z., and Veiga, M. d. 2019. A Regional Legacy Soil Dataset for Prediction of Sand and Clay Content with Vis-Nir-Swir, in Southern Brazil. Revista Brasileira De Ciencia Do Solo, 43.

[Figure]

Sorenson, P. T., Small, C., Tappert, M. C., Quideau, S. A., Drozdowski, B., Underwood, A., and Janz, A. 2017. Monitoring organic carbon, total nitrogen, and pH for reclaimed soils using field reflectance spectroscopy. Canadian Journal of Soil Science, 97(2), 241-248.

**Table 2.** Cross-validation results for soil organic carbon prediction using reflectance spectroscopy data.

| Model | Cross-validation results | | |
|---|---|---|---|
| | RMSE | $R^2$ | RPD |
| Multivariate adaptive regression splines | 0.66 | 0.76 | 2.0 |
| Artificial neural nets | 1.56 | 0.01 | 0.9 |
| Support vector machines | 0.67 | 0.75 | 2.0 |
| Partial least squares regression | 0.90 | 0.54 | 1.5 |
| Random forest | 0.62 | 0.78 | 2.1 |
| Cubist | 0.60 | 0.80 | 2.2 |

Note: Model results are evaluated based on the root mean square error (RMSE), $R^2$, and the ratio of performance to deviation (RPD).

**Table 4.** Cross-validation results for soil pH prediction using reflectance spectroscopy data.

| Model | Cross-validation results | | |
|---|---|---|---|
| | RMSE | $R^2$ | RPD |
| Multivariate adaptive regression splines | 0.54 | 0.58 | 1.5 |
| Artificial neural nets | 6.39 | 0.01 | 0.1 |
| Support vector machines | 0.51 | 0.60 | 1.6 |
| Partial least squares regression | 0.68 | 0.31 | 1.2 |
| Random forest | 0.47 | 0.67 | 1.7 |
| Cubist | 0.44 | 0.69 | 1.8 |

Note: Model results are evaluated based on the root mean square error (RMSE), $R^2$, and the ratio of performance to deviation (RPD).

**Table 3.** Cross-validation results for total nitrogen prediction using reflectance spectroscopy data.

| Model | Cross-validation results | | |
|---|---|---|---|
| | RMSE | $R^2$ | RPD |
| Multivariate adaptive regression splines | 0.06 | 0.75 | 2.1 |
| Artificial neural nets | 0.12 | 0.20 | 1.0 |
| Support vector machines | 0.06 | 0.78 | 2.1 |
| Partial least squares regression | 0.07 | 0.67 | 1.8 |
| Random forest | 0.06 | 0.78 | 2.1 |
| Cubist | 0.05 | 0.81 | 2.5 |

Note: Model results are evaluated based on the root mean square error (RMSE), $R^2$, and the ratio of performance to deviation (RPD).

**Table 5.** Cross-validation results from the Cubist model for samples collected from natural and reclaimed soils.

| Parameter | Cubist model cross-validation results | | |
|---|---|---|---|
| | RMSE | $R^2$ | RPD |
| Natural soil carbon | 0.60 | 0.84 | 2.5 |
| Reclaimed soil carbon | 0.59 | 0.70 | 1.8 |
| Natural soil nitrogen | 0.06 | 0.82 | 2.3 |
| Reclaimed soil nitrogen | 0.05 | 0.76 | 2.0 |
| Natural soil pH | 0.48 | 0.62 | 1.6 |
| Reclaimed soil pH | 0.39 | 0.68 | 1.8 |

Note: Model results are evaluated based on the root mean square error (RMSE), $R^2$, and the ratio of performance to deviation (RPD).

From both studies, Cubist > RF and SVM.

Several studies have also shown the superiority of CNN to SVM and other models for large dataset, for example, Tsakiridis et al. showed that CNN performs better than PLS, Cubist, SVM, and SBL algorithms. SVM has the same performance as Cubist.

Tsakiridis, N.L., Keramaris, K.D., Theocharis, J.B. and Zalidis, G.C., 2020. Simultaneous prediction of soil properties from VNIR-SWIR spectra using a localized multi-channel 1-D convolutional neural network. Geoderma, 367, p.114208.

Hence, we did not see the usefulness of repeating another comparison, which is not the aim of the paper. There are a lot of machine learning algorithms out there; including RT, RF, SVM,NN, ANN, MBL, etc. It is endless if we keep making comparison with every single model.

- In my opinion the section "sensitivity analysis" in mathodology does not add much to manuscript and in addition it is confusion and not clear, it needs to be removed or at least revised or shortend.

Response: As we mentioned in the manuscript, CNN is known as black box. As this is a soil science journal, we believe it is important to be able to interpret a complex model to understand how the decision was made by the model.

- The Conclusion is vague and requires revision and clarification.

Response:

We have clearly outlined our conclusion which answer the objectives of our paper. For sample size < 2000, the performance of CNN (with its current architecture) is not better than PLSR and Cubist. The increase of performance can only be seen when sample size > 2000. Thus we can recommend to other researchers that it would not be useful to try CNN unless you have enough data.